# Epidemiological modeling of SARS-CoV-2 in white-tailed deer (*Odocoileus virginianus*) reveals conditions for introduction and widespread transmission

Elias Rosenblatt[1]*, Jonathan D. Cook[2], Graziella V. DiRenzo[3,4], Evan H. Campbell Grant[5], Fernando Arce[4], Kim M. Pepin[6], F. Javiera Rudolph[2,7], Michael C. Runge[2], Susan Shriner[6], Daniel P. Walsh[8], Brittany A. Mosher[1]

1 Rubenstein School of Environment and Natural Resources, University of Vermont, Burlington, Vermont, United States of America, 2 U.S. Geological Survey, Eastern Ecological Science Center, Laurel, Maryland, United States of America, 3 U. S. Geological Survey, Massachusetts Cooperative Fish and Wildlife Research Unit, University of Massachusetts, Amherst, Massachusetts, United States of America, 4 Department of Environmental Conservation, University of Massachusetts, Amherst, Massachusetts, United States of America, 5 U.S. Geological Survey, Eastern Ecological Science Center, Turner's Falls, Massachusetts, United States of America, 6 National Wildlife Research Center, USDA, APHIS, Fort Collins, Colorado, United States of America, 7 Department of Ecosystem Sciences and Management, Pennsylvania State University, University Park, Pennsylvania, United States of America, 8 U. S. Geological Survey, Montana Cooperative Wildlife Research Unit, University of Montana, Missoula, Montana, United States of America

* erosenbl@uvm.edu

**Data Availability Statement:** All code and generated data used in this study are available in

## Abstract

Emerging infectious diseases with zoonotic potential often have complex socioecological dynamics and limited ecological data, requiring integration of epidemiological modeling with surveillance. Although our understanding of SARS-CoV-2 has advanced considerably since its detection in late 2019, the factors influencing its introduction and transmission in wildlife hosts, particularly white-tailed deer (*Odocoileus virginianus*), remain poorly understood. We use a Susceptible-Infected-Recovered-Susceptible epidemiological model to investigate the spill-over risk and transmission dynamics of SARS-CoV-2 in wild and captive white-tailed deer populations across various simulated scenarios. We found that captive scenarios pose a higher risk of SARS-CoV-2 introduction from humans into deer herds and subsequent transmission among deer, compared to wild herds. However, even in wild herds, the transmission risk is often substantial enough to sustain infections. Furthermore, we demonstrate that the strength of introduction from humans influences outbreak characteristics only to a certain extent. Transmission among deer was frequently sufficient for widespread outbreaks in deer populations, regardless of the initial level of introduction. We also explore the potential for fence line interactions between captive and wild deer to elevate outbreak metrics in wild herds that have the lowest risk of introduction and sustained transmission. Our results indicate that SARS-CoV-2 could be introduced and maintained in deer herds across a range of circumstances based on testing a range of introduction and transmission risks in various captive and wild scenarios. Our approach and findings will aid One Health strategies that mitigate persistent SARS-CoV-2 outbreaks in white-tailed deer populations and potential spillback to humans.

the R software package whitetailedSIRS (https://doi.org/10.5066/P9TZK938).

**Funding:** This work was supported by the Coronavirus Aid, Relief, and Economic Security Act (P.L. 116-136; ER, JDC, ECG, FA, MCR, and BAM). This research was supported in part by the U.S. Department of Agriculture, Animal and Plant Health Inspection Service (KMP and SS). The funders had no role in study design, data collection and analysis, decision to publish, or preparation of the manuscript.

**Competing interests:** The authors have declared that no competing interests exist.

## Author summary

Novel zoonotic diseases persist and evolve in both human and non-human hosts, posing challenges for human and animal health. SARS-CoV-2, the virus responsible for the COVID-19 pandemic in humans, has been detected in white-tailed deer (*Odocoileus virginianus*) yet we have limited understanding of its introduction and transmission within deer populations. Here, we use epidemiological models to describe SARS-CoV-2 introduction and transmission patterns within wild and captive white-tailed deer populations. We found that captive deer herds faced a higher risk of spillover of SARS-CoV-2 from humans and higher transmission rate compared to wild deer. Despite these differences in wild and captive contexts, we found that transmission in both wild and captive contexts often resulted in persisting circulation of SARS-CoV-2. This circulation was determined by deer-deer transmission in both wild and captive contexts, rather than by high rates of introduction from humans. Outbreaks in wild populations were greater when these animals could interact with captive deer populations circulating SARS-CoV-2. SARS-CoV-2 appears to circulate in both captive and wild deer across a range of circumstances. Our findings help to inform how to best mitigate the introduction and spread of SARS-CoV-2 in deer, with benefits to protecting human health.

## Introduction

Many emerging infectious diseases in animal populations are transmissible between animals and humans, representing a public health threat [1,2]. These diseases are called zoonoses and pose One Health challenges, meaning closely linked human, animal, and ecosystem health challenges that often require coordinated, multi-disciplinary action in the face of socioecological complexity and limited data [3,4]. Epidemiological models are powerful in understanding and responding to One Health challenges posed by zoonoses. Using the best-available science, epidemiological models can project the behavior of zoonotic disease spread across a range of possible conditions, quantify transmission risk between various host species, and examine the drivers influencing the introduction and transmission of zoonotic pathogens in wildlife hosts [5]. These exploratory inferences are particularly valuable with emerging infectious diseases and can complement monitoring efforts documenting the spatiotemporal distribution of infections [6,7].

Severe acute respiratory syndrome coronavirus 2 (SARS-CoV-2) in the subgenera *Sarbecoviruses*, subfamily *Orthocoronavirinae*, is a zoonotic virus that poses One Health challenges around the globe [8,9]. SARS-CoV-2 infection can result in severe respiratory disease (known as COVID-19) and death in humans, yet in wildlife species SARS-CoV-2 severity is highly variable. Since it was first documented in humans in late 2019, the number of known SARS-CoV-2 hosts has increased and includes a range of companion and wild animals, including wild and captive white-tailed deer (*Odocoileus virginianus*; hereafter deer)[10,11]. Transmission of SARS-CoV-2 can occur between humans, humans and animals, and between animals [12,13]. Each of these transmission pathways is concerning from a public health perspective for several reasons. First, SARS-CoV-2 circulating in human and non-human hosts can persist, recombine, and evolve into novel variants that change the properties of this pathogen [14–17]. Second, non-human hosts can act as a reservoir for SARS-CoV-2, posing risks of SARS-CoV-2 persisting outside of human hosts [18,19]. Lastly, SARS-CoV-2 may spill back to humans from non-human hosts as a potentially more virulent form of SARS-CoV-2 [12]. Collectively, these

concerns have given rise to surveillance programs of SARS-CoV-2 in wild and captive white-tailed deer across North America [20].

Two introduction pathways may have led to the transmission of SARS-CoV-2 from humans to deer, a process commonly referred to as 'spillover'. First, wild and captive deer could have been exposed to SARS-CoV-2 via direct interactions between humans and deer that are nearby. This direct pathway likely is a result of the aerosolized transmission of SARS-CoV-2 from humans to deer, given the tissue tropism in the upper respiratory tract of both species [21,22]. Direct interactions between humans and deer are possible in some areas of North America where deer are habituated to humans to the point where proximity or even contact is possible [23]. Human-deer interactions are also common in captive settings, ranging from facilities and herd management activities to exposition opportunities for visitors. Second, deer could have been exposed to SARS-CoV-2 indirectly through contaminated surfaces, feed, water, or through intermediate animal hosts [24]. While this indirect pathway has been postulated, evidence of transmission through this pathway does not currently exist.

Like SARS-CoV-2 spillover from humans to deer, the spread of SARS-CoV-2 within a white-tailed deer population could also occur via direct and indirect pathways. Transmission between deer could occur given various social interactions in wild and captive settings, including various agonistic and mating behaviors [25,26]. Direct transmission of SARS-CoV-2 between deer might include aerosolized and fluid transmission. Aerosolized transmission of SARS-CoV-2 between deer could occur within captive facilities where deer densities are high or in wild settings when deer are near one another. Fluid exchange could also lead to the transmission amongst deer given social behaviors such as allogrooming in seasonal social groups [27]. Indirect transmission of SARS-CoV-2 between deer may be possible through fomites, such as contaminated surfaces or feed, however, as previously mentioned, evidence of indirect transmission between deer is lacking.

Although our knowledge of SARS-CoV-2 has greatly increased over the last three years, factors influencing the introduction and transmission of SARS-CoV-2 in wildlife hosts and spillover risk remain poorly understood. Therefore, we develop a SIRS (Susceptible-Infected-Recovered-Susceptible) epidemiological model and apply it to wild and captive deer populations in a range of scenarios to address the following five objectives:

**Objective 1:** Evaluate human-deer (introduction) and deer-deer transmission (spread) in wild and captive deer scenarios to understand the role of pathways in disease dynamics;

**Objective 2:** Examine potential ranges of average prevalence, persistence, and incidence proportion of SARS-CoV-2 outbreaks in deer in wild and captive scenarios;

**Objective 3:** Understand the sensitivity of prevalence, persistence, and incidence proportion to introduction and spread across all scenarios;

**Objective 4:** Test if SARS-CoV-2 outbreaks in deer require continual introduction from humans or just a single introduction event;

**Objective 5:** Identify how contact between deer in captive and wild scenarios through fence line interactions can influence SARS-CoV-2 prevalence and persistence system-wide.

Collectively, this study provides insights into the dynamics of SARS-CoV-2 outbreaks in white-tailed deer populations and provides evidence for different mechanisms of spillover and persistence.

## Methods

### General approach and terms

We modeled SARS-CoV-2 transmission between humans and white-tailed deer, and among deer in several scenarios, including two types of captive facilities and wild deer in rural and suburban environments. We estimated direct (aerosolized) transmission rates from humans to deer as causing initial deer infections (human-to-deer, hereafter HtD). We estimated direct (aerosolized and fluid pathways) transmission rates within wild and captive deer populations following introduction from humans (deer-to-deer, hereafter DtD). We used these transmission rates to estimate two important epidemiological parameters (Objective 1). The introduction of a pathogen, such as SARS-CoV-2 into deer populations, can be quantified as the common Force-Of-Infection metric from humans to deer (FOI$_{HD}$; Fig 1) [28]. Then, SARS-CoV-2 transmission within a deer population can be quantified by the basic reproductive metric, $R_0$, or the number of new infections, in a completely naive population, originating from one infectious deer over the duration of its infection, with values greater than one indicating sustained infection throughout a population and values less than one indicating pathogen fade-out. (Fig 1, 28].

We projected the outbreak of infections across 120 days in each scenario to incorporate fall deer behavior (September-December). We focused on the fall season as deer reproductive behavior results in increased DtD contact rates and multiple hunting seasons and seasonal captive activities could increase HtD interactions. We used these fall projections to estimate the prevalence, persistence, and incidence proportion of SARS-CoV-2 in various types of simulated white-tailed deer populations (Fig 1; Objective 2). We used our simulated data to investigate the interaction between epidemiological parameters (introduction and transmission) and outbreak characteristics in deer populations (prevalence, persistence, and incidence

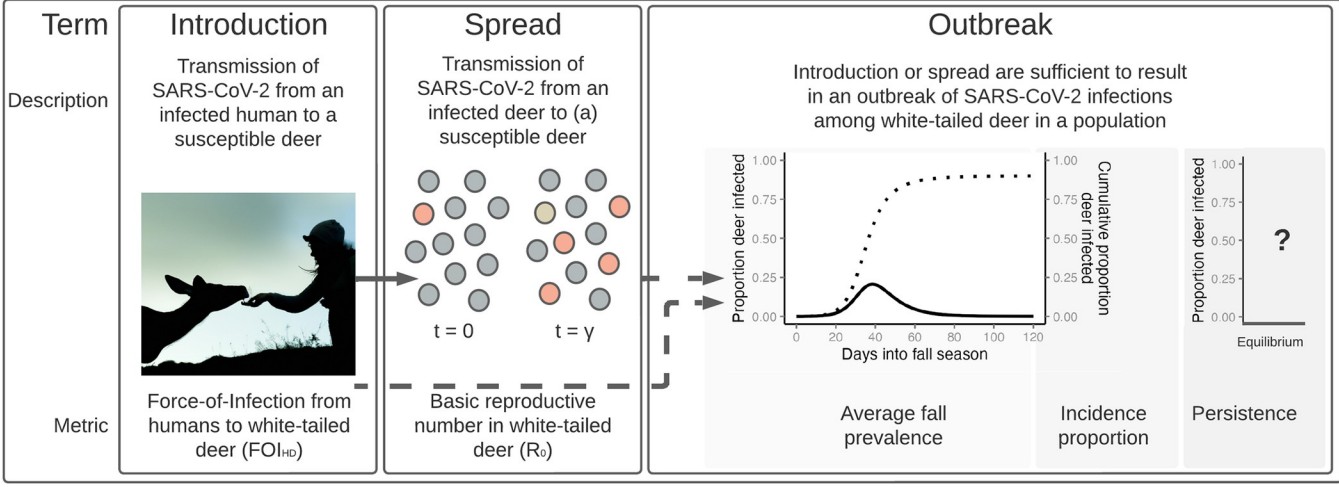

**Fig 1. The three stages of zoonotic spillover from humans to persistence in white-tailed deer.** In each stage outlined above, we describe the stage, illustrate the concept, and define the metric we use to characterize each stage across multiple scenarios of deer in wild and captive environments. We consider the introduction of SARS-CoV-2 into white-tailed deer populations through aerosolized transmission from an infected human, quantified as the Force-Of-Infection (FOI$_{HD}$). Transmission occurs as an infected deer (orange circle) interacts with susceptible deer (gray circles), transmitting SARS-CoV-2 through aerosols and fluid over the course of the animal's infectious period ($\gamma$). When the individual recovers from its infection (gold circle), it will have stemmed several secondary infections (orange circle), quantified as the basic reproductive number ($R_0 = 4$). Depending on the magnitude of FOI$_{HD}$ and $R_0$ (dashed arrows), an outbreak of infections may occur across a deer population. Average prevalence in the Fall season is averaged across daily values (dark line) and incidence proportion can be calculated through the projected fall season (dotted line). This outbreak will either persist or fade determined by the deterministic steady state of the set of ODE equations considered in this study, referred to here as equilibrium (x-axis). The image of a human hand-feeding a deer was created with the assistance of DALL-E 2.

proportion; Objective 3). We contrasted outbreak dynamics from continuous introduction from humans, compared to those from a single, initial infection event with no further introduction from humans (Objective 4). Finally, we ran the 120-day projection for wild and captive populations connected through a single-layer fence to explore how interactions between captive and wild deer may influence the prevalence and persistence of SARS-CoV-2 in both populations (Objective 5).

## Epidemiological model

To understand SARS-CoV-2 transmission between humans and deer and within deer populations, we developed a two-host (captive and wild deer) Susceptible-Infected-Recovered-Susceptible (SIRS) model (Fig 2, 5]. We considered two primary introduction pathways, including aerosolized SARS-CoV-2 transmission in shared airspace, and fluid transmission from sputum or other contagious discharges upon direct contact. For DtD transmission, we integrated both transmission pathways, while for HtD transmission, we estimated aerosolized transmission only. Humans were included as a source of infection, but human disease dynamics were not modeled as a response to disease dynamics in deer.

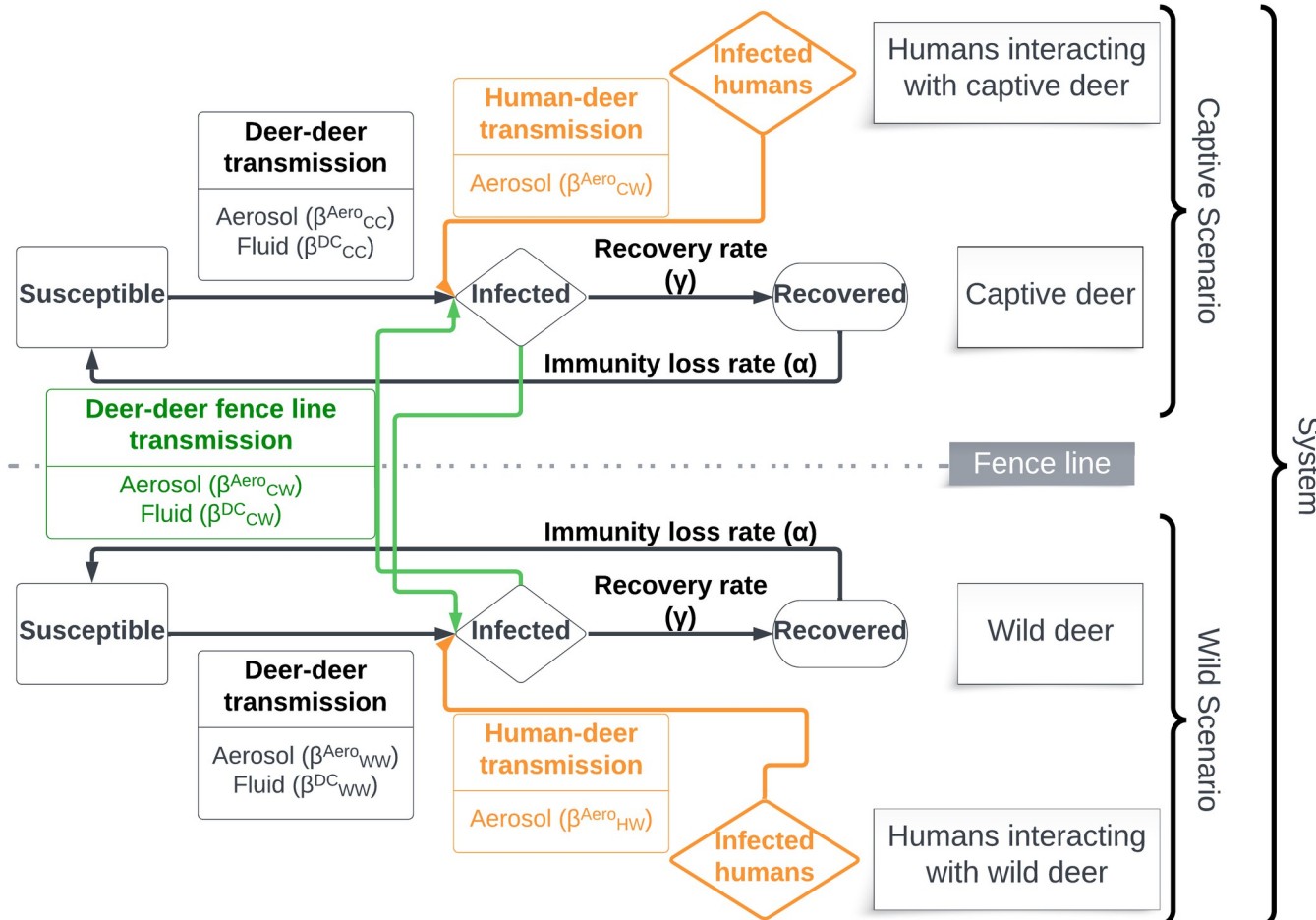

**Fig 2. A conceptual diagram of the Susceptible-Infectious-Recovered-Susceptible (SIRS) epidemiological model used for this simulation study.** Objectives that focused on specific captive or wild scenarios had no deer-deer fence line transmissions, preventing transmission between captive or wild populations. Objective 5 focused on how fence line transmission in captive-wild systems influence outbreak dynamics on both sides of the fence.

We made several assumptions either inherent in our SIRS approach or that incorporate patterns documented in the relevant literature. We assume that: transmission rates are additive; transmission rates are the same for naïve susceptible deer and recovered deer that have lost temporary immunity and are again susceptible; DtD transmission rates in wild scenarios and captive scenarios mimic wild conditions and are intermediate between frequency- and density-dependent transmission [29]. DtD transmission rates in intensive captive scenarios and across fence lines, and HtD transmission rates in all scenarios are constant and frequency-dependent, based on available data; DtD transmission rates via fluids only occurs when an infected and a susceptible individual are in proximity, including along fence lines; human prevalence is constant across each 120-day projection; there is homogenous mixing within captive and wild deer populations; recovery from infection and loss of immunity do not differ between captive and wild deer; there is no viral evolution; there is no disease-induced mortality [22]; there is no spillback from deer to humans (or at least, such spillback does not affect the disease dynamics in the deer population); and deer populations are closed, with no births, deaths, immigration, or emigration. On this last assumption, we recognize that many deer are harvested in the season we chose to simulate. We assume that harvest is random within the population such that the proportion of individuals within the various disease compartments of the SIRS model are unaffected.

The SIRS model was specified with a system of six ordinary differential equations (ODE) [5], and we derived rates for aerosolized and fluid transmission. We tracked the fractions of a population that are susceptible (s), infected (i), and recovered (r), rather than the number of individuals in each compartment. Human prevalence is fixed and not explicitly modeled in this study ($i_H$). In the equations that follow, our notation includes superscripts to indicate the mode of transmission, including: "Aero", to indicate transmission by aerosols; and "DC" to indicate transmission via fluid exchanged through direct contact. We use subscripts to indicate the individuals in a particular transmission interaction: transmission between wild deer (WW); transmission between captive and wild deer (CW); transmission between captive deer (CC); transmission from humans to wild deer (HW); and transmission from humans to captive deer (HC). We derived transmission rates as the product of HtD and DtD proximity rates and infection probabilities from previous studies. We used expert-elicitation to estimate any parameters unavailable in the literature. For more detail about parameter estimation, see the Scenario Descriptions section below.

## Ordinary differential equation

Three ODEs describe the disease dynamics in the wild deer population, with the change in the fraction of the wild population that is susceptible ($s_W$) given by

$$\frac{ds_W}{dt} = \alpha r_W - s_W\left(\beta_{WW}^{Aero}i_W + \beta_{WW}^{DC}i_W + \beta_{CW}^{Aero}i_C + \beta_{CW}^{DC}i_C + \beta_{HW}^{Aero}i_H\right),$$  (1)

the change in the fraction of the wild population that is infected ($i_W$) given by

$$\frac{di_W}{dt} = s_W\left(\beta_{WW}^{Aero}i_W + \beta_{WW}^{DC}i_W + \beta_{CW}^{Aero}i_C + \beta_{CW}^{DC}i_C + \beta_{HW}^{Aero}i_H\right) - \gamma i_W,$$  (2)

and the change in the fraction of the wild population that is recovered ($r_W$) given by

$$\frac{dr_W}{dt} = \gamma i_W - \alpha r_W,$$  (3)

where α is the immunity loss rate; β is the transmission rate specific to the infectious and

susceptible host recipient type (e.g., wild or captive deer) and interactions (i.e., aerosolized or direct contact); and $\gamma$ is the recovery rate from infection (Fig 2).

Three additional ODEs describe the disease dynamics in captive deer, with the change in the fraction of the captive population that is susceptible ($s_C$) given by

$$\frac{ds_C}{dt} = \alpha r_C - s_C\left(\beta_{CC}^{Aero} i_C + \beta_{CC}^{DC} i_C + \beta_{CW}^{Aero} i_W + \beta_{CW}^{DC} i_W + \beta_{HC}^{Aero} i_H\right), \tag{4}$$

the change in the fraction of the captive population that is infected ($i_C$) given by

$$\frac{di_C}{dt} = s_C\left(\beta_{CC}^{Aero} i_C + \beta_{CC}^{DC} i_C + \beta_{CW}^{Aero} i_W + \beta_{CW}^{DC} i_W + \beta_{HC}^{Aero} i_H\right) - \gamma i_C, \tag{5}$$

and the change in the fraction of the captive population that is recovered ($r_C$) given by

$$\frac{dr_C}{dt} = \gamma i_C - \alpha r_C. \tag{6}$$

We monitored proportions through these projections to reduce assumptions about population size in either wild or captive settings. We note that we summarized these continuous changes into discrete, daily S, I, and R compartment sizes for our analysis for ease of interpretation.

## Aerosolized transmission

Aerosolized transmission rates between a host $i$ and recipient $j$ ($\beta_{ij}^{Aero}$) can be described as

$$\beta_{ij}^{Aero} = \omega_{ij} * \sigma^{Aero} \tag{7}$$

where $\omega_{ij}$ is the proximity rate between host-recipient($i,j$) type (human-wild deer, human-captive deer, wild deer-wild deer, captive deer-captive deer, wild deer-captive deer, captive deer-wild deer); and $\sigma^{Aero}$ is the probability of infection from aerosols.

We define proximity $\omega_{ij}$ as the frequency per day that host $i$ and recipient $j$ are within 1.5 meters (m) of each other, drawn from existing social distancing guidelines for humans which range from 1–2 meters [30,31]. We estimate the proximity rate for wild deer, $\omega_{WW}$, based on a contact rate model developed by Habib et al. [32] for chronic wasting disease in white-tailed deer that permits density- or frequency-dependent transmission as well as intermediate cases that blend these two standard transmission processes. This rate applies to deer-deer transmission in most scenarios, including cases with and without attractants (e.g., bait, supplemental feed). We apply this model for captive circumstances that mimic natural conditions. It is given by

$$\omega_{ij} = \kappa\left(\frac{N_W^{(1-q)}}{A_W}\right) * \rho_{attractant} \tag{8}$$

where $\kappa$ is a scaling constant; q is a concavity scaling constant of the density-contact rate relationship ranging from 0–1, which allows an intermediate blend of density-dependence to frequency-dependence, respectively [32]; $N_W$ is the total population size; $A_W$ is the area inhabited by the population; $\rho_{attractant}$ is the adjustment for the presence of an attractant ($\rho_{attractant} = 1$ indicates no attractants present; $\rho_{attractant} > 1$ indicates attractants present).

All other proximity rates, including captive-captive deer ($\omega_{CC}$), captive deer-wild deer ($\omega_{CW}$), human-wild deer ($\omega_{HW}$), and human-captive deer ($\omega_{HC}$) were not explicitly modeled, and instead were drawn from parametric distributions.

The probability of infection, $\sigma^{Aero}$, given proximity, is a function of the instantaneous dose received and a Wells-Riley dose-response relationship given by

$$\sigma^{Aero} = 1 - e^{-\theta Q} \tag{9}$$

where $\theta$ is the species-specific rate of infection from 1 quantum of SARS-CoV-2; and Q is the dose (quanta) received by a single contact with an infected individual. Buonanno et al. [33] defines a quantum as "the dose of airborne droplet nuclei required to cause infections in 63% of susceptible human individuals." Therefore, $\theta > 1$ corresponds to 1 quantum causing infection in >63% of susceptible individuals, and $\theta < 1$ corresponds to 1 quantum causing infection in <63% of susceptible individuals [33–35].

To estimate the dose received by a susceptible individual (Q) we modeled (1) the emission of SARS-CoV-2 from an infectious individual ($ER_q$) and (2) the resulting concentration of SARS-CoV-2 in a designated airspace around an infectious individual, considering viral emission and viral loss. First, an infected individual emits virions at a particular rate ($ER_q$; quanta/hr) as the product of the viral load in its exhalation ($C_v$; RNA copies/ml), a conversion factor ($C_i$; quanta/RNA copy), the inhalation/exhalation rate (IR; m$^3$/hr), and the exhaled droplet volume concentration ($V_{drop}$; ml droplets/m$^3$ exhaled) [36] given by

$$ER_q = C_v * C_i * IR * V_{drop}. \tag{10}$$

We then use the emission rate to model the instantaneous concentration of virions (C; quanta/m$^3$) in a well-mixed airspace ($V_{air}$; m$^3$) around an infected individual ($ER_q$; quanta/hr). We assumed that the airspace around an infected individual was a half-sphere with a radius of 1.5 m, or 7.07 m$^3$. We account for viral loss as the sum of air exchange (AER; hr$^{-1}$), settling (s; hr$^{-1}$), and inactivation ($\lambda$; hr$^{-1}$)[33]. Thus, the instantaneous concentration is given by

$$C = \frac{ER_q}{(AER + s + \lambda) * V_{air}}. \tag{11}$$

When a susceptible individual enters the contaminated airspace surrounding an infectious individual, the dose (Q; quanta) is the product of the inhalation rate of the susceptible individual (IR; m$^3$/hr), the concentration of virions in the fixed volume (C; quanta/m$^3$), and the duration of contact ($t_{contact}$; hr) given by

$$Q = IR * C * t_{contact}. \tag{12}$$

## Fluid transmission

We model fluid transmission rate for deer conditional on proximity with another deer (Eq 8). Fluid transmission rates between a host and recipient ($\beta_{ij}^{DC}$) are given by

$$\beta_{ij}^{DC} = \omega_{ij} * \varepsilon^{DC} * \sigma^{DC} \tag{13}$$

where $\omega_{ij}$ is the proximity rate between host-recipient(ij) type (wild deer-wild deer, captive deer-captive deer, captive deer-wild deer); $\varepsilon^{DC}$ is the probability of direct contact conditional on proximity; and $\sigma^{DC}$ is the probability of infection from direct contact.

The probability of infection, $\sigma^{DC}$, given contact, was modeled similarly to Eq 9, as a log-logistic function of dose and the reciprocal probability of infection given exposure to a single dose, k [37]. The dose received is a product of the transferred sputum volume given contact, $V_{sputum}$, and viral concentration in sputum, $C_v$ given by

$$\sigma^{DC} = 1 - e^{-((C_v \times V_{sputum})/k)} \tag{14}$$

where $C_v$ is the viral concentration in sputum (in plaque-forming units; PFU); $V_{sputum}$ is the volume of sputum transferred given contact; and k is the reciprocal of the probability of a single PFU causing infection.

## Scenario descriptions

We estimated HtD and DtD transmission and outbreak characteristics in four scenarios: (1) wild deer in a rural setting, (2) wild deer in a suburban setting, (3) captive deer in an outdoor ranch, and (4) captive deer in an intensive facility (Fig 2). These scenarios span a range of possible habitat or captive facility conditions, deer densities, and proximity rates with humans; although each of these variables is a continuous metric, we discretized the scenarios to make them easier to interpret.

Below, we present parameter estimates used in each simulation (Table 1). For parameters that were unavailable in the literature, we conducted expert elicitation using the IDEA protocol and a four-point elicitation process [38,39]. We included 11 experts on two separate panels: one focused on SARS-CoV-2 virology and another on deer behavior in captive and wild settings. The estimates for 13 parameters we solicited from experts are listed in Table 1. Elicitation methods, the elicitation questions for each panel, and individual (anonymous) and aggregated probability distributions are reported in S1 and S2 Files and S1–S13 Figs. For study Objectives 1 to 4, fence line transmission was fixed at zero to capture outbreak dynamics within these specific scenarios. This transmission rate was restored for the final study objective exploring the influence of linked scenarios across fence lines in outbreak dynamics.

*Wild deer in a rural setting*–Wild deer are free-ranging in an area with a rural human density (3.1 humans/km$^2$; 15$^{th}$ percentile of U.S. counties with <100 humans/km$^2$ overlapping white-tailed deer range; [45–47]. We assumed that deer interacted with humans during regulated hunting either using still-hunting, or ground blind or tree stand tactics but were not harvested. We also assumed that baiting and backyard feeding were illegal but may still occur. We calculated wild DtD proximity rates using a population density of 10 deer/km$^2$ for an area with 26% wooded habitat [32]. For aerosol transmission, we assumed that proximity rates for deer approaching within 1.5m of each other were equal to Habib et al.'s [32] estimated proximity rate of deer approaching within 25m of each other. HtD transmission was derived by estimating the rate and duration of human-deer proximity events and a fixed human prevalence of 5% (Table 1). Wild deer in a rural setting had the lowest rate and duration of these human-deer proximity events (Table 1). We calculated and applied air-exchange rates (AER; 4$^{-hr}$) based on a 15-minute residence time drawn from a range of published values for forest airflow studies (Table 1)[43,44].

*Wild deer in a suburban setting*–Wild deer are free-ranging in an area of suburban human density (100 humans/km$^2$)[45]. DtD proximity rates were derived using the same parameters as used in the rural scenario, and the AER value used was the same as in the rural scenario (Table 1). Wild deer in a suburban setting experience higher HtD transmission rates, driven by higher HtD proximity rates and longer duration of proximity events, relative to wild deer in a rural setting (Table 1).

*Captive deer in an outdoor ranch*–We considered captive deer in an outdoor ranch facility typical of a managed, fenced hunting reserve. We assumed that deer stocking densities resulted in the same DtD proximity rates as were estimated in wild scenarios, with an increase in proximity rates due to supplemental feeding (Table 1). We used the same AER value as in wild settings as these captive individuals reside outside. We assume HtD proximity rates are the same as those estimated for the "wild deer in a suburban setting" scenario, but the typical duration of these proximity events is longer in this scenario, reflecting those typical of a captive facility (Table 1).

**Table 1. Model parameter estimates for SARS-CoV-2 Susceptible-Infected-Recovered-Susceptible (SIRS) ordinary differential equations (ODE).**

| Equations | Variable | Definition (units) | Captive | | Wild | | Source |
|---|---|---|---|---|---|---|---|
| | | | Outdoor ranch | Intensive facility | Rural | Suburban | |
| 1, 3, 4, 6 | α | Immune loss rate (day$^{-1}$; log-normal) | μ = 4.72, σ = 0.63 | | | | This study, expert elicited |
| 2, 3, 5, 6 | γ | Recovery rate (day$^{-1}$) | 1/6 days | | | | [21] |
| 1, 2, 4, 5 | $I_H$ | Human Prevalence (%) | 5% | | | | Assumed and fixed |
| 8 | κ | Proximity rate scaling adjustment (unitless) | 11.35 | NA | 11.35 | 11.35 | [32] |
| 8 | q | Proximity rate concavity scaling constant (unitless) | 0.34 | NA | 0.34 | 0.34 | [32] |
| 8 | $N_w$ | Number of deer per unit area ($A_w$) | 1000 | NA | 1000 | 1000 | [32] |
| 8 | $A_w$ | Area for intermediate density-dependence (km$^2$) | 100 km$^2$ | NA | 100 km$^2$ | 100 km$^2$ | [32] |
| 8 | $\rho_{attractant}$ | Adjustment for the presence of an attractant (bait, feed, etc.; log-normal) | μ = 3.47, σ = 0.23 | NA | NA | NA | This study, expert elicited |
| - | $\omega_{HW}$ | Human-deer proximity rate (events/120 days; log-normal) | μ = 0.57, σ = 0.95 | μ = 2.52, σ = 1.13 | μ = -1.59, σ = 1.70 | μ = 0.572, σ = 0.951 | This study, expert elicited |
| - | $\omega_{CC}$ | Deer proximity rate in captivity (events/day; log-normal) | NA | μ = 3.47, σ = 0.91 | NA | NA | This study, expert elicited |
| - | $\omega_{WC}$ | Wild-captive deer proximity rate along fences (events/day, only included for Objective 4) | 0.00072 direct contacts/day / $\sigma^{DC}$ | | | | [40,41] |
| 9 | θ | Quanta SARS-CoV-2 dose-response in deer (1/quanta required for ID63; log-normal) | μ = 0.28, σ = 0.27 | | | | This study, expert elicited |
| 10 | $C_i$ | Conversion from SARS-CoV-2 RNA copies to quanta | 0.0014 quantum/RNA copy | | | | [36] |
| 10 | $C_v$—human | Concentration of SARS-CoV-2 in human sputum (RNA copies/ml) | μ = 5.6 log10 RNA copies/ml, σ = 1.2 log10 | | | | [33] |
| 10,14 | $C_v$—deer | Concentration of SARS-CoV-2 in deer sputum (RNA copies/ml; log-normal) | μ = 0.22, σ = 0.34; proportional to $C_v$—human | | | | This study, expert elicited |
| 10 | IR—human | Inhalation rate for humans, standing (m$^3$/hr) | 0.53 m$^3$/hr | | | | [36] |
| 10, 12 | IR—deer | Inhalation rate for deer, breathing (m$^3$/hr) | 0.85 m$^3$/hr | | | | [42] |
| 10 | $V_{drop}$ | Droplet volume concentration (speaking; ml/m$^3$) | 0.01 ml/m3 | | | | [36] |
| 11 | $V_{air}$ | Volume of shared airspace with 1.5m radius (m$^3$) | 7.07 m$^3$ | | | | This study, calculated |
| 11 | AER | Air exchange rate ($^{-hr}$) | 4$^{-hr}$ | 1$^{-hr}$ | 4$^{-hr}$ | 4$^{-hr}$ | [43,44] |
| 11 | s | SARS-CoV-2 settling rate ($^{-hr}$) | 0.24$^{-hr}$ | | | | [33] |
| 11 | λ | SARS-CoV-2 inactivation rate ($^{-hr}$) | 0.63$^{-hr}$ | | | | [33] |
| 12 | $t_{contact}$ | Duration of proximity event between human and deer (minutes; log-normal) | μ = 1.79, σ = 1.15 | | μ = -0.36, σ = 0.98 | μ = 0.432, σ = 0.929 | This study, expert elicited |
| 12 | $t_{contact}$ | Duration of proximity event between deer (all proximity types; minutes; log-normal) | μ = 1.55, σ = 1.27 | | | | This study, expert elicited |
| 13 | $\epsilon^{DC}$ | Probability of deer making direct contact (logit-normal) | μ = -1.46, σ = 0.71 | | | | This study, expert elicited |
| 14 | k | Dose-response function for plaque-forming units (PFU required for ID63) | 410 | | | | [37] |
| 14 | $V_{sputum}$ | Volume of sputum transferred between individuals on contact (μl) | 100 μl | | | | Fixed |

Equations refer to in-line equation numbers. Mean and standard deviation (μ and σ), along with error distribution are listed for expert-elicited estimates (S1–S13 Figs). Parameters which do not apply to particular scenarios are indicated (NA).

*Captive deer in an intensive facility*–The last scenario considered was captive deer in a captive breeding or exposition facility. Deer in this type of facility were predominantly indoors at high stocking densities and low indoor air exchange rates (AER; $1^{-hr}$). Both DtD and HtD proximity rates and duration were highest in this scenario (Table 1).

*Objective 1*: *Differences in human-to-deer and deer-to-deer transmission across scenarios*–We quantified the strength of HtD transmission in each scenario using Force-Of-Infection calculations from humans to deer ($FOI_{HD}$; Eq 15). These FOI calculations are based on HtD transmission rates ($\beta_{HD}^{Aero}$; Eq 7) and human prevalence ($i_H$) and equate to the proportion of susceptible deer infected by infectious humans per day.

$$FOI_{HD} = \beta_{HD}^{Aero} i_H \qquad (15)$$

We also report the probability of at least one HtD transmission per 1,000 deer ($N$) over the fall season (t = 120 days), using a constant hazard model (Eq 16)[48].

$$p(HtD|FOI_{HD}, N, t) = 1 - (e^{-FOI_{HD}t})^N \qquad (16)$$

We quantified the strength of DtD transmission for each scenario using the number of susceptible deer infected by a single infectious deer, $R_0$, derived from the sum of aerosol and fluid transmission rates over the recovery period from infection ($\gamma$; Eq 17). Again, $R_0$ values greater than one indicate sustained transmission throughout a population, and values less than one indicate pathogen fade-out.

$$R_0 = \frac{\beta_{ij}^{Aero} + \beta_{ij}^{DC}}{\gamma}. \qquad (17)$$

We compared $FOI_{HD}$, *p(HtD)*, and $R_0$ estimates across scenarios to evaluate differences in the potential for SARS-CoV-2 to be transmitted from humans to deer and then spread amongst deer. All calculations were conducted in R [49]. We summarized the sensitivity of $FOI_{HD}$ and $R_0$ to expert-elicited parameters (S14 and S15 Figs). We focused on expert-elicited parameters for these sensitivities as these parameters had the greatest uncertainty in our calculations. We did not present sensitivity of *p(HtD)* to expert-elicited parameters as *p(HtD)* was derived from FOI.

*Objective 2*: *Average prevalence, persistence of infection, and incidence proportion in each scenario*–We used the six ODEs for the SIRS model, parameters estimated from the literature or expert elicitation, and derived transmission parameters to project continual SARS-CoV-2 introduction and spread across each scenario of interest (Table 1). From these projections, we calculated the proportion of individuals in the wild, captivity, or in both settings that were susceptible, infectious, or recovered. We ran 1,000 iterations for each of the four scenarios. Each iteration had a randomly drawn parameter set, where we randomly drew one value from each parameter distribution during each iteration, resulting in 1,000 parameter sets used to project outbreaks in each scenario (Table 1). Parameters that were constant across scenarios did not vary between parameter sets which ensured that any observed variation was due to differences across scenarios, and not sampling variation from repeated random draws from error distributions.

We projected the proportional size of each SIRS compartment for 120 days for each iteration, using the ODE solver *ode()* from the deSolve package in R [49,50]. We estimated the average daily prevalence of deer in each scenario during the 120-day projection. We determined if SARS-CoV-2 would persist beyond the 120-day projection for each iteration using the *runsteady()* function from the rootSolve package [51,52] to estimate the deterministic stable state from the SIRS ODE equation. We assigned each iteration a logical value if infectious

compartment at equilibrium was >0.1% for each iteration (at least 1 deer infected out of 1 000). We estimated mean probability of persistence and 95% binomial confidence intervals using the *binom.confint()* function with the exact method from the binom package for each scenario [53]. Finally, we tracked the incidence proportion, or cumulative proportion of the population infected over the 120 days during these simulations for wild and captive deer (Eqs 18 and 19). This incidence proportion could exceed 1, indicating that all individuals in the population were infected at least once.

$$Incidence\ proportion_W$$
$$= \sum_{t=1}^{120} s_{W,t-1}(\beta_{WW}^{Aero} i_{W,t-1} + \beta_{WW}^{DC} i_{W,t-1} + \beta_{CW}^{Aero} i_{C,t-1} + \beta_{CW}^{DC} i_{C,t-1} + \beta_{HW}^{Aero} i_H) \tag{18}$$

$$Incidence\ proportion_C$$
$$= \sum_{t=1}^{120} s_{C,t-1}(\beta_{CC}^{Aero} i_{C,t-1} + \beta_{CC}^{DC} i_{C,t-1} + \beta_{CW}^{Aero} i_{W,t-1} + \beta_{CW}^{DC} i_{W,t-1} + \beta_{HC}^{Aero} i_H) \tag{19}$$

We summarized these three measures across iterations in each scenario with the median value and 80% confidence intervals. These include median average prevalence, median probability of persistence, and median incidence proportion.

*Objective 3*: *Sensitivity of prevalence, persistence and incidence proportion to spillover and spread*–We tested the sensitivity of prevalence, persistence, and incidence proportion of SARS-CoV-2 in white-tailed deer to different levels of spillover (FOI) and spread ($R_0$). After each iteration, we categorized outcomes by one of the following spread categories: unsustained spread ($R_0 < 1$), low, sustained spread ($1 < R_0 \leq 3$), medium, sustained spread ($3 < R_0 \leq 5$), and high, sustained spread ($R_0 > 5$). We used the *stat-smooth()* function from the ggplot2 package [54] to visualize trends between HtD transmission, as quantified by FOI, and outbreak metrics for each spread category.

*Objective 4*: *SARS-CoV-2 outbreaks in deer from a single introduction event*–We tested whether a SARS-CoV-2 outbreak can occur following a single spillover event, in contrast to the continual introduction modeled above for the other objectives. We simulated this introduction as an initial event that resulted in 0.1%, 1e-4%, and 1e-7% prevalence in deer at the start of the 120-day projection, with no further introduction from humans. We compared differences in prevalence, persistence, and incidence proportion between these initial spillover simulations and the continuous spillover simulation investigated for the other objectives.

*Objective 5 Effects of fence line interactions between wild and captive deer on SARS-CoV-2 prevalence and persistence on either side of the fence*–We extended our SIRS model to allow fence line interactions between captive and wild deer. To do this we projected outbreaks for paired captive -wild scenarios separated by a fence, using combinations of the two captive and two wild scenarios and associated parameters described above (n = 4 combinations; hereafter systems). We added fence line contact probability and allowed all individuals to interact along fence lines, enabling proximity and direct contact (Table 1).

## Results

### Objective 1: Differences of introduction and spread for white-tailed deer across settings

The risk of introduction of SARS-CoV-2 from humans to deer varied within and across scenarios (Eqs 15 and 16, respectively; Fig 3 and Table 2). Median $FOI_{HD}$ estimates were 1244-, 85-, and 19-times higher in the intensive facility, outdoor ranch, and wild deer in suburban scenarios, respectively, relative to median $FOI_{HD}$ estimates for rural, wild deer (Table 2).

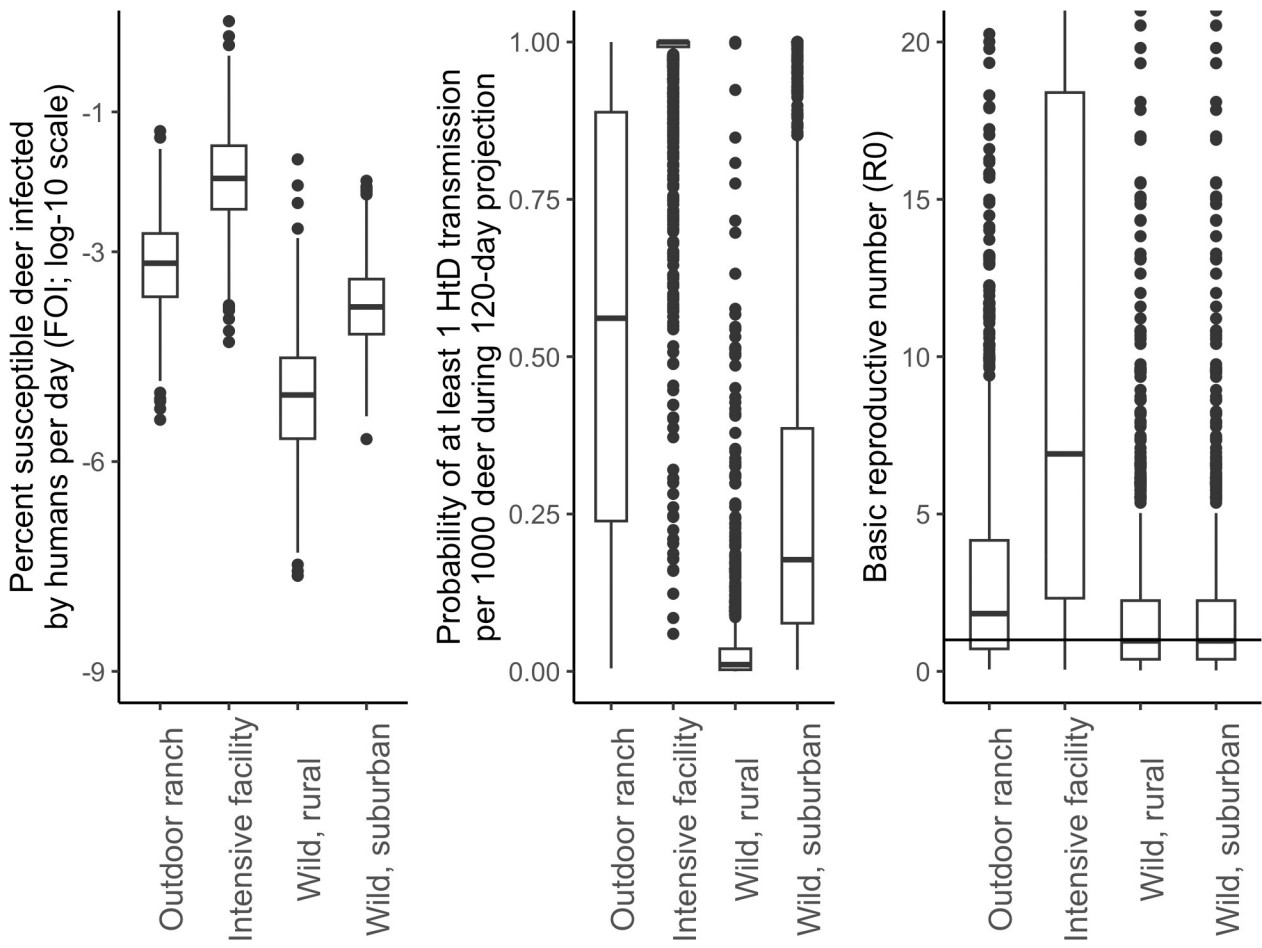

**Fig 3. Variation in Force-Of-Infection from humans-to-deer (FOI), probability of at least 1 human-to-deer (HtD) transmission, and basic reproductive numbers ($R_0$) across the four scenarios considered in this study.** Human Force-Of-Infection is log10 transformed and presented as odds of HtD transmission per deer, per day. The basic reproductive number threshold between unsustained and sustained transmission from deer-to-deer is indicated with a horizontal line ($R_0 = 1$). Box plots depict the minimum, first quartile, median, third quartile, and maximum, with outliers depicted as single points.

**Table 2. Median metrics and 80% confidence intervals for simulated SARS-CoV-2 outbreaks in white-tailed deer in four scenarios.**

| Scenario | FOI_HD | p(HtD, 1:1000) | $R_0$ | Average prevalence | Persistence | Incidence proportion |
|---|---|---|---|---|---|---|
| Intensive facility | 0.0112% | 100.0% | 6.91 | 7.2% | 95% | 150% |
| | (0.0017–0.0915%) | (86.4–100.0%) | (0.84–43.15) | (0.39–11.7%) | (94.0–96.0%) | (8.0–245%) |
| Outdoor ranch | 0.0007% | 56.1% | 1.83 | 4.2% | 69% | 85% |
| | (1e-4–0.005%) | (9.7–99.7%) | (0.31–8.83) | (0.003–8.8%) | (65.9–71.5%) | (0.06–183%) |
| Wild, rural | <0.0001% | 1.1% | 0.97 | 0.001% | 47% | 0.03% |
| | (0–0.0001%) | (0.1–11.2%) | | (0–6.6%) | (43.9–50.2%) | (0–138%) |
| Wild, suburban | 0.0002% | 17.7% | (0.17–4.36) | 0.01% | 49% | 0.30% |
| | (0–0.001%) | (3.5–66.3%) | | (4e-4–6.9%) | (45.6–51.9%) | (0.01–142%) |

Metrics include: the proportion of susceptible deer infected by humans, per day (Force-Of-Infection from humans-to-deer, FOI_HD); the probability of at least 1 in 1,000 deer becoming infected from a human during the fall season (probability of human-to-deer transmission, p(HtD, 1:1,000)); the number of susceptible deer infected by an infected deer ($R_0$); the average daily prevalence during the fall season (average prevalence); the probability of SARS-CoV-2 persisting beyond the simulated fall season (Persistence); and the total proportion of the population infected during the fall season (incidence proportion).

$FOI_{HD}$ was highly sensitive to the frequency and duration of proximity between humans and deer (S14). Median probabilities of at least one HtD transmission per 1000 deer ranged from 100%, 56.1%, 17.7%, and 1.1% in the intensive facility, outdoor ranch, wild suburban, and wild rural scenarios, respectively (Table 2). There was high uncertainty around risk of introduction in each scenario, with detectable differences between the intensive facility and wild deer in rural setting using 80% confidence intervals (Table 2).

SARS-CoV-2 transmission between deer ($R_0$; Eq 18) was greater in captive scenarios relative to wild scenarios, with most iterations sustaining transmission of SARS-CoV-2 among the deer population (Table 2). Transmission in both wild scenarios were nearly identical, with 51.3% of iterations resulting in $R_0$ values too small to sustain transmission of SARS-CoV-2 ($R_0$ <1; median $R_0$ = 0.97; Table 2). $R_0$ values were highly variable in each scenario leading to no detectable differences with 80% confidence (Table 2). $R_0$ was sensitive to several parameters, including duration of a deer-deer proximity event, the concentration of SARS-CoV-2 in deer sputum, and SARS-CoV-2 dose-response in deer (S15). $R_0$ in captive, intensive facilities was sensitive to deer-deer proximity rate due to the uncertainty around the aggregate estimate from expert elicitation (S15).

## Objective 2: Average prevalence, persistence of infection, and incidence proportion in each setting

Simulated outbreaks of SARS-CoV-2 were variable across scenarios, with higher average prevalence, probability of persistence, and incidence proportion in captive scenarios relative to wild scenarios (Table 2 and Fig 4). Intensive facilities had the highest average prevalence, probability of SARS-CoV-2 persistence, and incidence proportion, followed by the outdoor ranch scenario and both wild scenarios (Table 2). Median outbreak metrics in both wild scenarios, while much lower than captive scenarios, were slightly elevated in the suburban setting compared to the rural setting (Table 2). Overall, there was high variability in these metrics in each scenario, with non-overlapping 80% confidence for the probability of persistence in the intensive facility, outdoor ranch, and wild scenarios (Table 2 and Fig 4).

## Objective 3: Sensitivity of prevalence, persistence and incidence proportion to spillover and spread

When we partitioned the relationship between $FOI_{HD}$ and outbreak characteristics, we found evidence that sensitivity to $FOI_{HD}$ differs depending on how quickly SARS-CoV-2 transmits ($R_0$, Fig 5). When deer-deer transmission is too low to sustain SARS-CoV-2 infections ($R_0$ <1), high $FOI_{HD}$ is required for non-zero average prevalence and incidence proportion during the projection, and for a high probability of infections persisting (Fig 5). As deer-deer transmission reaches self-sustaining levels (1< $R_0$ <3), the role of $FOI_{HD}$ has a greater influence on average prevalence, persistence, and incidence proportion (Fig 5). As $R_0$ continues to increase to medium (3< $R_0 \leq$ 5) and high spread ($R_0$ > 5), the sensitivity of prevalence and incidence proportion to $FOI_{HD}$ diminishes, and persistence is no longer sensitive to changes in $FOI_{HD}$. (Fig 5).

## Objective 4: SARS-CoV-2 outbreaks in deer from a single introduction event

Differences in outbreak characteristics exist between continual introduction of SARS-CoV-2 from humans and a single, initial introduction (Fig 6). However, these differences vary depending on the size of the initial introduction and the scenario and uncertainty prevented

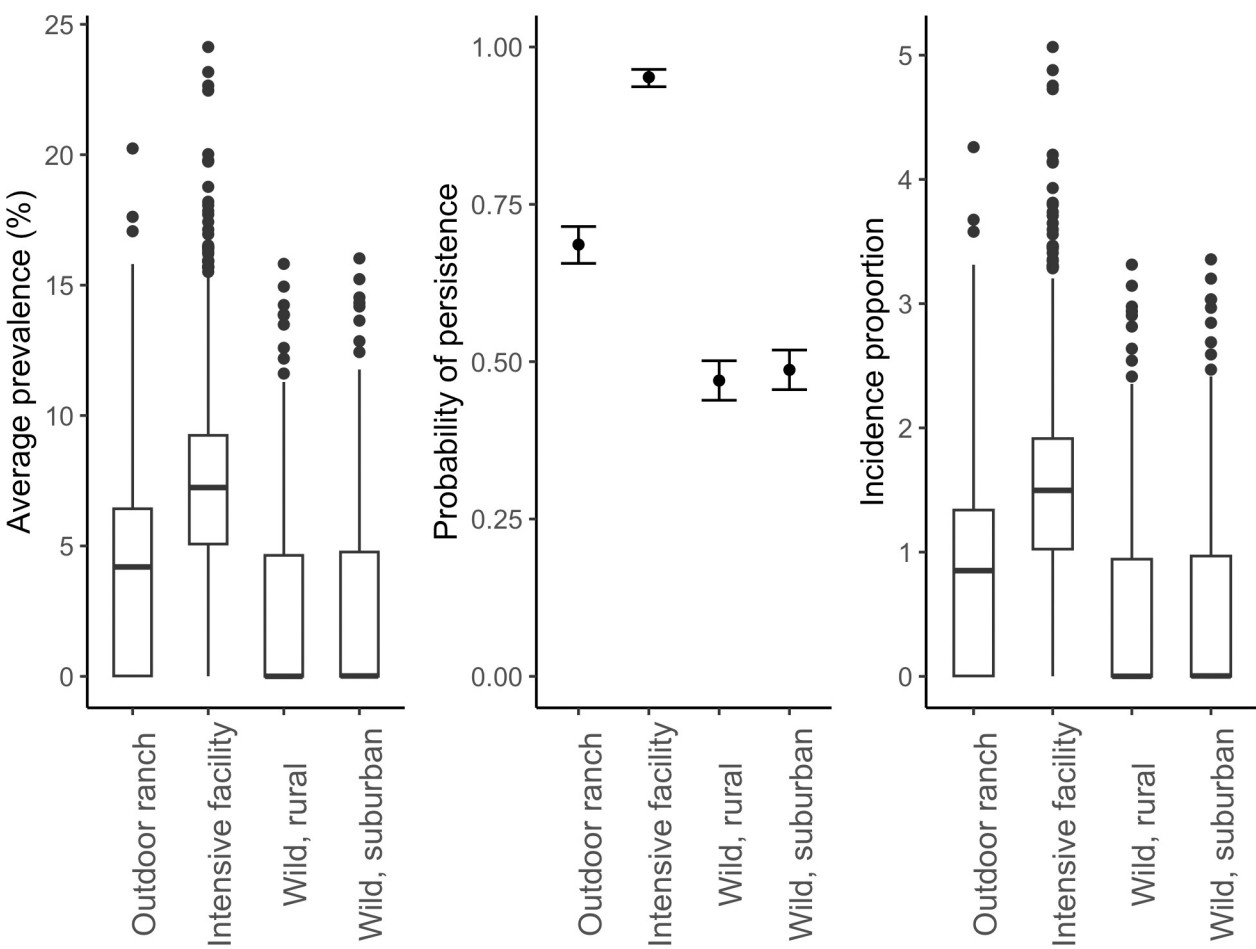

**Fig 4. Distributions of average prevalence, persistence probability, and incidence proportion values during the 120-day fall projection in each scenario of interest. 1000 simulations were run for each scenario**Box and whisker plots depict the minimum, first quartile, median, third quartile, and maximum, with outliers depicted as single points. Error bars for persistence represent 95% confidence intervals.

high confidence in these differences. If an initial, single introduction resulted in 0.1% prevalence in any context, the average prevalence and incidence proportion were slightly greater than the average prevalence and incidence proportion when SARS-CoV-2 was continuously introduced. However, probability of persistence decreased in all scenarios except for wild deer in a rural setting, where probability of persistence would increase with this initial prevalence compared to when SARS-CoV-2 was continuously introduced. With an initial prevalence of 0.0001%, all scenarios showed median average prevalence and incidence proportion similar to or slightly lower than when SARS-CoV-2 was continuously introduced. The probability of persistence was consistent with those estimated for an initial 0.1% prevalence. Finally, with an initial prevalence of 1e-7%, the lowest tested, all scenarios showed decreases in average prevalence, probability of persistence, and incidence proportion relative to other continuous or initial infection conditions. However, even at this low level of initial infection, deer in the intensive facility scenario had median average prevalence and median incidence proportion that were comparable to when SARS-CoV-2 was continuously introduced, albeit with greater variability.

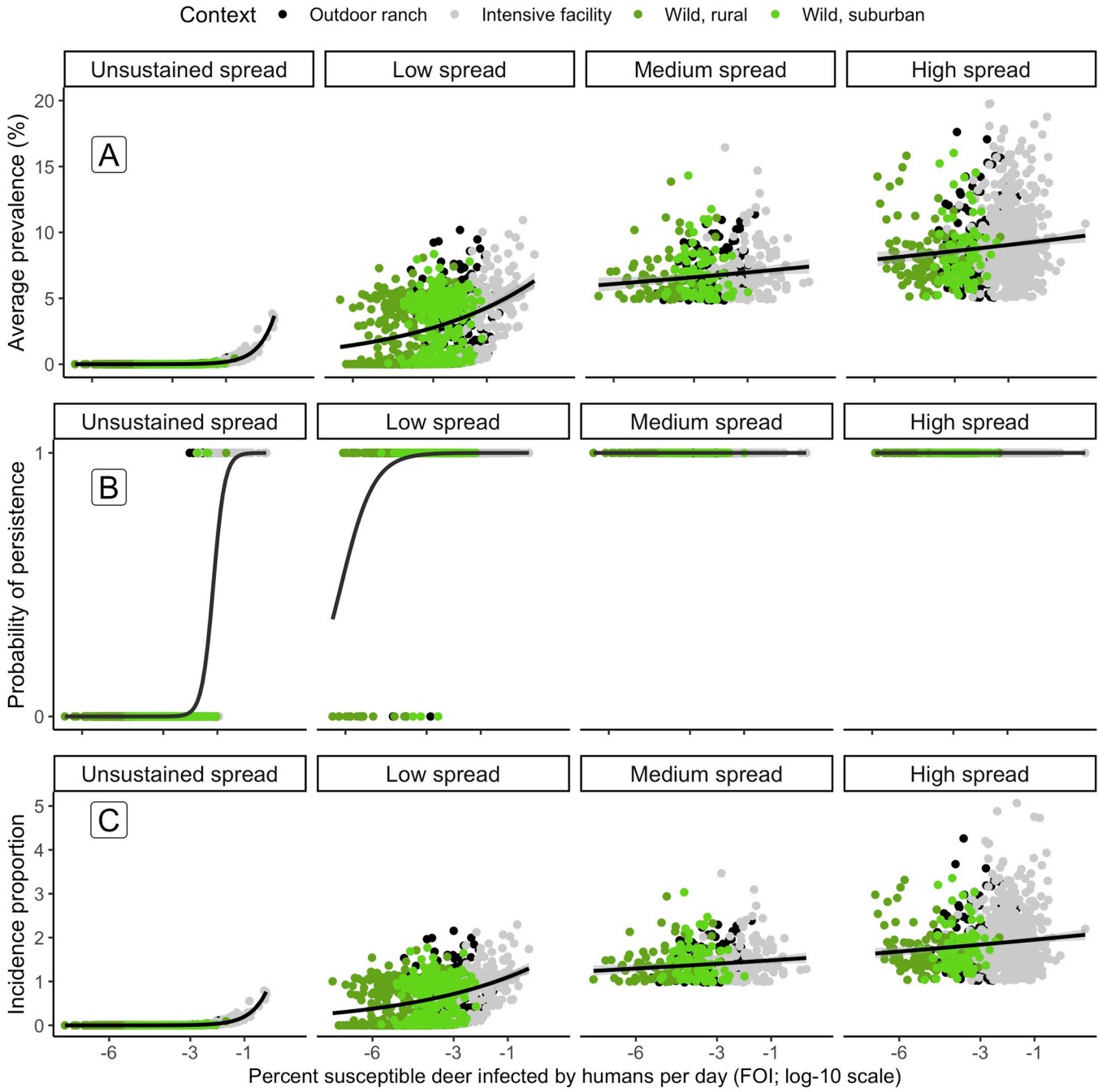

**Fig 5. The relationship between human-to-deer Force-Of-Infection and (A) average SARS-CoV-2 prevalence, (B) persistence of SARS-CoV-2, and (C) the incidence proportion during the fall, dependent on the degree of transmission from deer-to-deer ($R_0$).** Points indicate metrics for each iteration simulated, with point color and shading indicating a particular scenario. Fitted lines indicate trends in the data, fitted with a log-normal or logistic-regression for prevalence and persistence, respectively. Transmission categories included unsustained transmission ($R_0 < 1$), low, sustained transmission ($1 < R_0 \leq 3$), medium, sustained transmission ($3 < R_0 \leq 5$), and high, sustained transmission ($R_0 > 5$).

## Objective 5: Effects of fence line interactions between wild and captive deer on SARS-CoV-2 prevalence and persistence on either side of the fence

When fence line interactions occurred between all combinations of captive and wild scenarios, wild deer had a higher prevalence and incidence proportion of SARS-CoV-2 during the fall

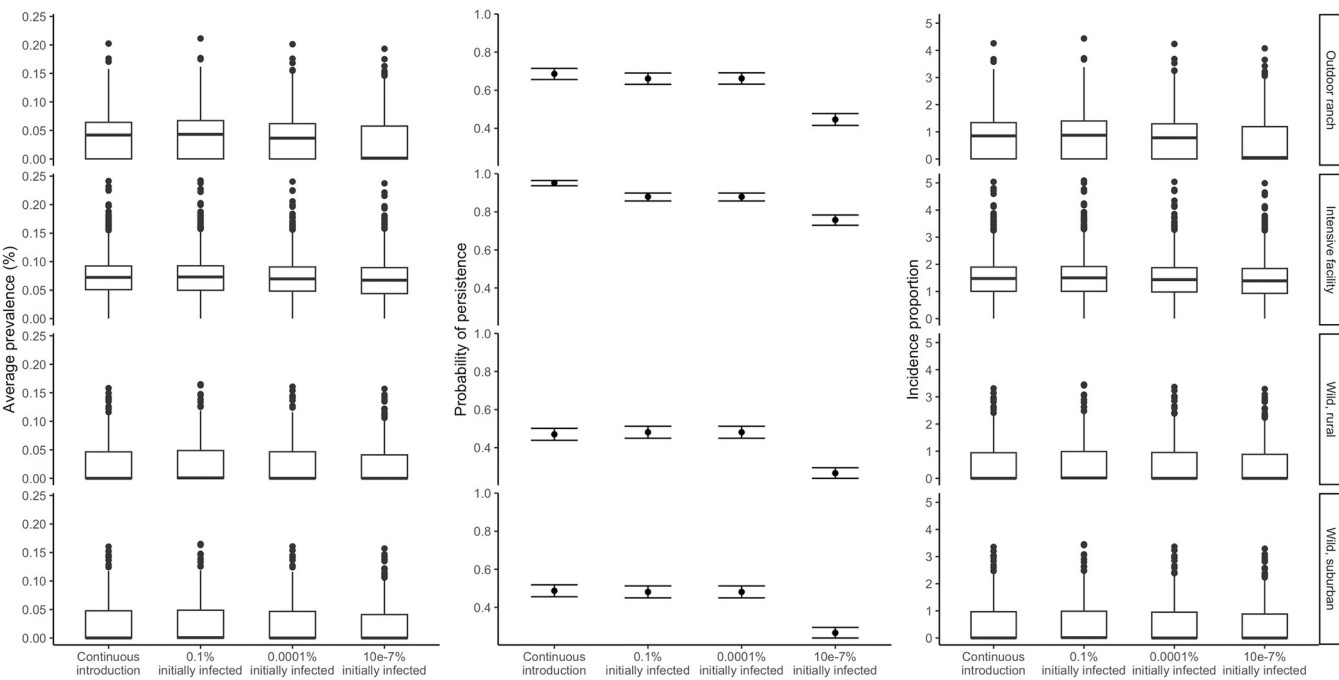

**Fig 6. Variation of average prevalence, persistence, and incidence proportion during the 120-day fall projection.** Error bars for persistence represent 95% confidence intervals. Plots are faceted by scenario, with variation in outbreak characteristics displayed for continuous introduction from humans, and various degrees of initial, single introductions with no continuous introduction from humans. Box plots depict the minimum, first quartile, median, third quartile, and maximum, with outliers depicted as single points.

projection compared to simulations without fence line interactions (Objective 2; Table 3). These increases were highly variable depending on the captive and wild conditions. The probability for persistence did not increase for wild deer when fence line interactions occurred, and captive deer did not experience an increase in any metric (Table 3). Of the four systems, fence line interactions had the greatest effect when dividing captive deer in an intensive facility and wild deer in a rural setting. In this system during the 120-day projection, the average prevalence in wild deer increased by approximately 122% (median), and the incidence proportion of the wild deer in a rural setting increased from 1e-5 to 0.278 (median, Table 3). Smaller

**Table 3. Increases in prevalence, persistence, and incidence proportion of SARS-CoV-2 outbreaks with simulated systems with deer in captive and wild scenarios interacting across fence lines.**

| System | Median increase in prevalence (80% CI) | | Median proportional increase in prevalence (80% CI) | | Mean increase in probability of persistence (80% CI) | | Median increase in incidence proportion (80% CI) | |
|---|---|---|---|---|---|---|---|---|
| | **Wild** | **Captive** | **Wild** | **Captive** | **Wild** | **Captive** | **Wild** | **Captive** |
| Outdoor ranch and wild, suburban | 0.002 (0–0.143) | 0 (0-8e-4) | 0.46% (0.01–104%) | 0.0028% (3e-4-0.0294) | 0.001 (1e4-0.004) | 0 | 0.044 (0–3.21) | 5e-4 (0–0.016) |
| Intensive facility and wild, suburban | 0.013 (0–0.554) | 0 (0-1e-4) | 15.37% (0.38–562%) | <1e-04% (0–0.0013) | 0.006 (0.003–0.011) | 0 | 0.231 (7e-4-11.62) | 0 (0–0.002) |
| Outdoor ranch and wild, rural | 0.004 (0–0.557) | 0 (0-8e-4) | 4.47% (0.08–3094%) | 9e-04% (0–0.019) | 0.015 (0.010–0.021) | 0 | 0.081 (0–10.95) | 1e-4 (0–0.016) |
| Intensive facility and wild, rural | 0.016 (0–1.105) | 0 (0-1e-4) | 184.35% (1.17–20655%) | 0% (0–0.001) | 0.019 (0.014–0.026) | 0 | 0.301 (7e-4-21.72) | 0 (0–0.001) |

CI = confidence interval.

increases were estimated in the intensive facility and wild deer in a suburban system (Table 3). We estimated similar patterns when considering systems with fence line interactions between outdoor ranch facilities and wild deer, albeit smaller in magnitude (Table 3).

## Discussion

Our study demonstrates the potential for variable, yet widespread risk of SARS-COV-2 introduction and spread across white-tailed deer populations in North America. Our findings indicated that epidemiological conditions and the proximity rates of white-tailed deer may lead to sustained transmission. We estimated sustained infections in wild and captive populations across a wide range of Force-Of-Infection rates from both continual spillover from humans and an initial spillover event. We also demonstrated that wild deer may experience higher prevalence, persistence, and incidence proportion of SARS-CoV-2 infections when sharing a fence line with captive facilities. These results complement ongoing, retrospective surveillance efforts across a range of captive and wild contexts by revealing the spillover risk of SARS-CoV-2 from infected humans and the risk of transmission between deer [20, 55]. More broadly, our approach provides a framework for using epidemiological modeling to evaluate the risks of outbreaks and sustained infections of SARS-CoV-2 and other zoonotic diseases in wildlife hosts in a variety of contexts.

Despite lower risks of introduction and transmission, SARS-CoV-2 was still able to transmit and sustain itself in wild scenarios. If $R_0$ was less than one, indicating unsustainable transmission, our two wild scenarios did not have sufficient $FOI_{HD}$ to sustain infections. However, when $R_0$ increased above one, wild scenarios showed rapid increases in average prevalence and incidence proportion, and a high probability of SARS-CoV-2 persisting into the future. Our findings generally match those reported by Hewitt et al. [55], who used surveillance data from wild deer across the United States of America to estimate infection rates and prevalence, and estimated $R_0$ greater than 1 in most of counties monitored across 27 states. In short, our results indicate that there may be broad circumstances where wild deer populations could face repeated introduction and sustained transmission of SARS-CoV-2.

Both captive scenarios showed a higher risk of introduction and a higher rate of transmission, resulting in higher prevalence and persistence relative to wild scenarios. Our findings conform to the available literature on the introduction and transmission of SARS-CoV-2 in captive populations. Roundy et al. [56] reported 94.4% seropositivity for one captive herd and 0% seropositivity in two other captive herds, one of which housed axis (*Axis axis*) and fallow deer (*Dama dama*). This contrast could indicate a difference in transmission from humans, as stocking conditions may increase the transmission of the virus. Our study also indicated different epidemiological dynamics in systems where captive and wild deer may interact through fence lines compared to systems without these interactions. However, despite the vulnerabilities of captive conditions to rapid transmission of SARS-CoV-2, we emphasize that the patterns of outbreaks in facilities and increased risk of fence line transmission are likely to vary through space and time. Our captive scenarios did not focus on single facilities with a particular herd size, but rather a pool of captive individuals. Introduction and transmission within individual facilities may be so rapid that a localized infection results in SARS-CoV-2 running out of susceptible hosts and the outbreak extinguishing itself. Spillover to wild populations through fence line interactions during localized outbreaks remain a risk for these individual facilities, though the risk of spillover from wild to captive facilities appears low.

White-tailed deer encounter a wide range of conditions across North America making it challenging to capture this variability in a single analysis. The four scenarios evaluated here are indicative of processes typical of both wild and captive conditions. Our analysis focused on

temporal patterns of SARS-CoV-2 introduction and spread across wild and captive white-tailed deer, yet spatial variation undoubtedly plays a role. We did not make our simulations spatially explicit, as we felt that our global approach met our objectives to better understand infection dynamics across typical conditions. Additionally, integrating a spatial component to this study would require specific spatial conditions and assumptions that either generalize across large geographic extents, or limit inferences to conditions in a specific locality. We feel these are important next steps given our inferences from this study and will aid in our understanding of the reported spatial and temporal heterogeneities of SARS-CoV-2 cases in white-tailed deer [10,19,24,57].

We were required to make several assumptions in our parameterization of the SIRS models that may have influenced our inferences. First, we used Watanabe et al.'s [37] reported infection probability for SARS-CoV in mice by intranasal exposure to estimate transmission of SARS-CoV-2 through fluid when deer make physical contact. We join other simulation studies that use this parameter estimate to calculate direct contact probability through fluid transfer and acknowledge the uncertainty of this parameter given it has not been quantified in the literature [58]. Second, we used the stable-state equilibrium of the SIRS model to infer the persistence of SARS-CoV-2. We acknowledge that this assumes that parameter values are not stochastic and do not change past the simulated fall season. Seasonal changes in white-tailed deer behavior are well-documented and affect introduction and spread for multiple pathogens in deer, as with other host-pathogen systems [59–61]. Third, parameters used to derive transmission risk between deer in our simulations did not vary by sex. Ongoing monitoring of SARS-CoV-2 in wild white-tailed deer populations indicate higher infection probability and seropositivity in male white-tailed deer, likely driven by sex-specific behaviors [55,62]. We believe that our inferences are robust with our integration of uncertainty around derived parameter estimates and the patterns of prevalence and persistence values documented in multiple studies monitoring ongoing infections [17].

Despite a growing number of studies of SARS-CoV-2 in white-tailed deer, there is no consensus on how SARS-CoV-2 is introduced into deer populations. This is a key detail in mitigating the introduction and transmission of SARS-CoV-2 in a prolific wildlife species that can interact with humans in both wild and captive contexts. In this study, an initial outbreak had to infect less than 10e-7% of deer for there to be an observable decrease in average prevalence, probability of persistence, and incidence proportion compared to those observed during continual spillover. These results indicate that an initial introductory event, even at a low rate, could result in an outbreak in both captive and wild settings. While introduction through aerosolized transmission from humans to deer is presumed to be most probable, our findings indicate that indirect sources of infection could play a role through a single transmission event. Infection from contaminated fomites or wastewater could initiate an outbreak given sufficient dose received by an individual. However, further research remains into the risk posed by these sources.

Sustained SARS-CoV-2 infections in this prolific wildlife species frequently interacting with humans in captive and wild settings creates a One Health challenge that affects human, animal, and ecosystem health. SARS-CoV-2 has demonstrated its ability to spread in wild and captive white-tailed deer populations across much of North America. The outbreak dynamics reported in this study indicate the ease by which the virus can be introduced and sustained in this non-human species. Surveillance studies indicate that multiple lineages of SARS-CoV-2 have been introduced and broadly circulated in white-tailed deer populations [10,13,19], with evidence of spillback from deer to humans [14,63]. Our modeling approach provides a foundation to evaluate risks to human, animal, and ecosystem health posed by zoonotic diseases, and to test potential interventions to meet this and other One Health challenges.

## Supporting information

**S1 File. Expert elicitation methods.**
(DOCX)

**S2 File. Descriptions of settings for Deer Ecology panel expert elicitation.**
(DOCX)

**S1 Fig. Responses by experts on the Virology panel to Question 1 to estimate immunity loss rate ($\alpha$).** Consider a healthy individual white-tailed deer that was recently infected with SARS-CoV-2 and has since recovered (i.e., the individual is no longer shedding infectious virions). Assume that, with recovery, this individual is temporarily immune to reinfection by SARS-CoV-2 if they were to be exposed to the virus by a dose otherwise sufficient to cause infection. However, after some period, this individual will lose their immunity and become susceptible to infection again. After how many days can this individual deer be reinfected with SARS-CoV-2 after it has fully recovered from an infection? (A) fitted log-normal probability distributions for answers provided by individual experts, and (B) the aggregated log-normal distribution of answers across experts (black line). The aggregate log-normal distribution has a median of 112.6 days (80% confidence interval: 50.5–251.4 days; grey range along x-axis).
(TIF)

**S2 Fig. Responses by experts on the Virology panel to Question 2 to estimate viral load in deer sputum ($C_{v\text{-}deer}$).** Consider a white-tailed deer infected with SARS-CoV-2 from which you can collect a sputum sample. What is the ratio of the average viral load of a deer compared to the average viral load in an infected human's sputum sample? (A) fitted log-normal probability distributions for answers from individual experts, and (B) the aggregated log-normal distribution of answers across experts. The aggregate log-normal distribution has a median viral load of 1.24 that found in humans (80% confidence interval: 0.80–1.93; grey range along x-axis). The vertical lines in both (A) and (B) refer to a ratio of 1, that corresponds to no difference between viral loads in deer and human sputum.
(TIF)

**S3 Fig. Responses by experts on the Virology panel to Question 3 to estimate aerosolized dose-response relationship for deer and SARS-CoV-2 ($\theta$).** For humans, $\theta = 1$, corresponding to a 1 quantum dose successfully infecting 63% of susceptible individuals, or HID63. Based on your expertise and knowledge of the literature, what do you expect the r value to be for the average, healthy white-tailed deer? (A) fitted log-normal probability distributions for answers from individual experts, and (B) the aggregated log-normal distribution of answers across experts. The aggregate log-normal distribution has a median dose-relationship of 1.32 (80% confidence interval: 0.93–1.87; grey range along x-axis). The vertical line indicates the human dose-response relationship, $\theta = 1$.
(TIF)

**S4 Fig. Responses by experts on the Deer Ecology panel to Question 4 to estimate the duration of deer staying in proximity of each other ($<1.5m$; $t_{contact}$)—Given that two individual deer are in proximity (within 1.5 m of each other), how long do you expect these individuals to stay in proximity on average (minutes)?** (A) fitted log-normal probability distributions for answers from individual experts, and (B) the aggregated log-normal distribution of answers across experts. The aggregate log-normal distribution has a median duration of 4.72 minutes (80% confidence interval: 0.93–24.11 minutes; grey range along x-axis).
(TIF)

**S5 Fig. Responses by experts on the Deer Ecology panel to Question 5 to estimate the probability of direct contact given proximity ($\varepsilon^{DC}$).** Given that two deer are in proximity (within 1.5m of each other), what is the probability that these individuals make direct contact? (A) fitted logit-normal probability distributions for answers from individual experts, and (B) the aggregated logit-normal distribution of answers across experts. The aggregate logit-normal distribution has a median direct contact probability of 0.19 (80% confidence interval: 0.09–0.37; grey range along x-axis).
(TIF)

**S6 Fig. Responses by experts on the Deer Ecology panel to Question 6 to estimate the influence of baiting or supplemental feeding ($\rho_{attractant}$).** If an individual deer has 17 proximity events with other deer each day in the absence of baiting, how many proximity events do you expect an individual deer to have with other wild deer in the presence of an attractant (bait, food, or other product intended to attract deer)? (A) fitted log-normal probability distributions for answers from individual experts, and (B) the aggregated log-normal distribution of answers across experts. The aggregate log-normal distribution has a median 32.2 proximity events per day when an attractant is present (80% confidence interval: 24.1–43.0). Relative to 17 proximity events in the absence of baiting, this aggregate distribution estimates an increase in proximity events by 1.90-fold when an attractant is present (80% confidence interval: 1.42–2.53; grey range along x-axis).
(TIF)

**S7 Fig. Responses by experts on the Deer Ecology panel to Question 7 to estimate the rate of a deer in proximity to a human in a rural setting ($\omega_{HW-rural}$).** Given the conditions outlined above [S2 File], how many times do you expect an individual deer to come into proximity with a human during the fall months (1 September– 31 December)? (A) fitted log-normal probability distributions for answers from individual experts, and (B) the aggregated log-normal distribution of answers across experts. The aggregate log-normal distribution has a median of 0.20 proximity events per deer, per fall in a rural setting (80% confidence interval: 0.02–1.80; grey range along x-axis).
(TIF)

**S8 Fig. Responses by experts on the Deer Ecology panel to Question 8 to estimate the duration of deer staying in proximity of a human in a rural setting (<1.5m; tcontact-HW, rural).** Given that a human and a deer come into proximity in a rural setting (within 1.5m of each other), how long do you expect these individuals (human and deer) to stay in proximity on average (minutes)? (A) fitted log-normal probability distributions for answers from individual experts, and (B) the aggregated log-normal distribution of answers across experts. The aggregate log-normal distribution has a median proximity duration of 0.70 minutes in a rural setting (80% confidence interval: 0.20–2.46 minutes; grey range along x-axis).
(TIF)

**S9 Fig. Responses by experts on the Deer Ecology panel to Question 9 to estimate the rate of a deer in proximity to a human in a suburban setting ($\omega_{HW-suburban}$).** Given the conditions outlined above [S2 File], how many times do you expect an individual deer to come into proximity with a human during the fall months (1 September– 31 December)? (A) fitted log-normal probability distributions for answers from individual experts, and (B) the aggregated log-normal distribution of answers across experts. The aggregate log-normal distribution has a median of 1.77 proximity events per deer, per fall in a suburban setting (80% confidence interval: 0.52–6.00; grey range along x-axis).
(TIF)

**S10 Fig. Responses by experts on the Deer Ecology panel to Question 10 to estimate the duration of deer staying in proximity of a human in a suburban setting (<1.5m; tcontact-HW, suburban).** Given that a human and a deer come into proximity in a suburban setting (within 1.5m of each other), how long do you expect these individuals (human and deer) to stay in proximity on average (minutes)? (A) fitted log-normal probability distributions for answers from individual experts, and (B) the aggregated log-normal distribution of answers across experts. The aggregate log-normal distribution has a median proximity duration of 1.54 minutes in a suburban setting (80% confidence interval: 0.47–5.07 minutes; grey range along x-axis).
(TIF)

**S11 Fig. Responses by experts on the Deer Ecology panel to Question 11 to estimate the rate of a deer in proximity to a human in an intensive captive setting ($\omega_{HC}$).** How many times do you expect an individual deer in a captive facility to come into proximity with a human (within 1.5m of each other) during the fall months (1 September– 31 December)? (A) fitted log-normal probability distributions for answers from individual experts, and (B) the aggregated log-normal distribution of answers across experts. The aggregate log-normal distribution has a median of 12.44 proximity events per deer, per fall in an intensive captive setting (80% confidence interval: 2.92–53.08; grey range along x-axis).
(TIF)

**S12 Fig. Responses by experts on the Deer Ecology panel to Question 12 to estimate the duration of deer staying in proximity of a human in a intensive captive setting (<1.5m; tcontact-CW).** Given that a human and a deer come into proximity in a captive setting (within 1.5m of each other), how long do you expect these individuals (human and deer) to stay in proximity on average (minutes)? (A) fitted log-normal probability distributions for answers from individual experts, and (B) the aggregated log-normal distribution of answers across experts. The aggregate log-normal distribution has a median proximity duration of 5.98 minutes in an intensive captive setting (80% confidence interval: 1.36–26.16 minutes; grey range along x-axis).
(TIF)

**S13 Fig. Responses by experts on the Deer Ecology panel to Question 13 to estimate the rate of a deer in proximity to another deer in an intensive captive setting (<1.5m; $\omega_{CC}$).** Given these captive conditions, how many times do you expect an individual deer to be in proximity with another deer in a day on average (within 1.5m of each other)? (A) fitted log-normal probability distributions for answers from individual experts, and (B) the aggregated log-normal distribution of answers across experts. The aggregate log-normal distribution has a median proximity rate of 32.15 events per day in an intensive captive setting (80% confidence interval: 9.97–103.61 events per day; grey range along x-axis).
(TIF)

**S14 Fig. Sensitivity of Force-Of-Infection (FOI) to expert-elicited parameters.** Each row corresponds to a scenario, and each column corresponds to a parameter included in the calculation of FOI. Points indicate each iteration's draw of each parameter and resulting derived parameter (FOI), with a trend line fitted to summarize the sensitivity of FOI to the range of drawn parameter values. Point and error bars on the top of each plot indicate the mean and 95% confidence intervals of the aggregate parameter distribution from expert elicitation.
(TIF)

**S15 Fig. Sensitivity of basic reproductive number (R0) to expert-elicited parameters.** Each row corresponds to a scenario, and each column corresponds to a parameter included in the calculation of $R_0$. Points indicate each iteration's draw of each parameter and resulting derived parameter ($R_0$), with a trend line fitted to summarize the sensitivity of $R_0$ to the range of drawn parameter values. Point and error bars on the top of each plot indicate the mean and 95% confidence intervals of the aggregate parameter distribution from the expert elicitation exercise. Deer-deer proximity rates for captive, outdoor ranch, wild, suburban, and wild, rural scenarios were drawn from Habib et al.'s (2014) [32] contact rate model, with less uncertainty compared to expert-elicited proximity rates in the captive, intensive facility scenario.
(TIF)

## Acknowledgments

We thank Paul Cross, Anna Fagre, Kate Huyvaert, Jeff Root, Jeff Chandler, Sarah Hamer, Kamen Campbell, Chris Jennelle, Jonathan Trudeau, Kurt Vandegrift, and Noelle Thompson for their participation in our expert elicitations. Any use of trade, firm, or product names is for descriptive purposes only and does not imply endorsement by the U.S. Government. This is publication #05 of the Disease Decision Analysis and Research Group (DDAR) of the U.S. Geological Survey.

## Author Contributions

**Conceptualization:** Elias Rosenblatt, Jonathan D. Cook, Graziella V. DiRenzo, Evan H. Campbell Grant, Kim M. Pepin, Michael C. Runge, Susan Shriner, Daniel P. Walsh, Brittany A. Mosher.

**Data curation:** Elias Rosenblatt, Fernando Arce, F. Javiera Rudolph.

**Formal analysis:** Elias Rosenblatt, Fernando Arce, F. Javiera Rudolph.

**Funding acquisition:** Brittany A. Mosher.

**Investigation:** Elias Rosenblatt, Jonathan D. Cook, Graziella V. DiRenzo, Evan H. Campbell Grant, Fernando Arce, Kim M. Pepin, F. Javiera Rudolph, Michael C. Runge, Susan Shriner, Daniel P. Walsh, Brittany A. Mosher.

**Methodology:** Elias Rosenblatt, Jonathan D. Cook, Graziella V. DiRenzo, Evan H. Campbell Grant, Fernando Arce, Kim M. Pepin, Michael C. Runge, Susan Shriner, Daniel P. Walsh, Brittany A. Mosher.

**Project administration:** Jonathan D. Cook, Graziella V. DiRenzo, Evan H. Campbell Grant, Michael C. Runge, Brittany A. Mosher.

**Resources:** Elias Rosenblatt, Kim M. Pepin, F. Javiera Rudolph, Susan Shriner, Daniel P. Walsh.

**Software:** Elias Rosenblatt, Fernando Arce, F. Javiera Rudolph.

**Supervision:** Jonathan D. Cook, Graziella V. DiRenzo, Evan H. Campbell Grant, Michael C. Runge, Brittany A. Mosher.

**Validation:** Elias Rosenblatt.

**Visualization:** Elias Rosenblatt.

**Writing – original draft:** Elias Rosenblatt.

**Writing – review & editing:** Elias Rosenblatt, Jonathan D. Cook, Graziella V. DiRenzo, Evan H. Campbell Grant, Fernando Arce, Kim M. Pepin, F. Javiera Rudolph, Michael C. Runge, Susan Shriner, Daniel P. Walsh, Brittany A. Mosher.

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
