## [Decision Letter · Decision Letter 0]

3 Feb 2024

Dear Mr. Rosenblatt,

Thank you very much for submitting your manuscript "Epidemiological modeling of SARS-CoV-2 in white-tailed deer (Odocoileus virginianus) reveals conditions for introduction and widespread transmission" for consideration at PLOS Computational Biology.

As with all papers reviewed by the journal, your manuscript was reviewed by members of the editorial board and by several independent reviewers. In light of the reviews (below this email), we would like to invite the resubmission of a significantly-revised version that takes into account the reviewers' comments.

We cannot make any decision about publication until we have seen the revised manuscript and your response to the reviewers' comments. Your revised manuscript is also likely to be sent to reviewers for further evaluation.

Sincerely,

Benjamin Althouse

Academic Editor

PLOS Computational Biology

Rob De Boer

Section Editor

PLOS Computational Biology

Reviewer's Responses to Questions

**Comments to the Authors:**

Reviewer #1: This manuscript is an interesting exploration of the spillover dynamics of SARS COVID19 from humans into white-tailed deer. I thought the expert elicitation exercise was an interesting way to obtain estimates of rates that are needed to parameterize multiple routes of spillover.

I am skeptical that the expert elicitation combined with literature defined rates of fine-scale infectivity and viral parameters from model systems is able to make the pathways identifiable. That said, I think the conclusions that are centered on that results that it takes a very little amount of spillover from humans can seed transmission are valuable.

Sensitivity and expert elicitation:

I think that the exploration of parameter space was sufficiently justified via the expert elicitation and scenario set-up. However, there is also little preventing more thorough exploration of the sensitivity of the objective outputs in broader parameter space. Particularly, Objective 1: In reading the methods, my initial thought was that this would be an ideal place to perform more thorough sensitivity analyses on the effect of the parameter uncertainty because its an analytical solution. Then line 510 in results describes variation in the FOI calculations. Was this a sensitivity analysis? Or incorporated uncertainty from the expert elicitation? How that variation in the FOI solutions needs to be included in the methods.

Additional comments are focused on clarifications of methods and interpretations:

Line 228- Clarify what you are deriving the transmission rates from up front. It will be very helpful to give a short explanation that you are deriving the rates by explicitly parameterizing contact and infectiousness, here, because this is where you first introduce this method. I was confused by how you were deriving these until I read through the methods several times.

Line 240 and parameter table: As I understand your methods, this integrates over continuous time so there are no daily change. I suggest cleaning up the explanation of parameter units to indicate that the rates are expressed in units that have a more natural interpretation, but are all transformed to be on the equivalent scale to be integrated over your 120 day time period

Line 392-393: Clarify this sentence. I don’t understand what the 2 proximities are in this assumption

Line 520-521: The statement that the R0 values with median 0.97 were too small to sustain transmission seems like too much interpretation for the results. Particularly when the range you report in Table 2 appears to include many R0 values >>1. I suggest adding a text description of the variation in R0 values in this section, in addition to presenting the median values.

Line 560-561: remove the text “deer infected by infected deer” in the parentheses. It’s confusing and you already have defined your interpretation of R0 for readers.

-line and page numbering stopped near the end of the results and I’ll do what I can to orient the remaining comments-

Discussion sentence beginning “Our findings indicated…”

You did not include any aspect of sociality. Some aspects were including in how you modeled transmission according to previous studies, but these were constant, as far as I could tell. Remove this conclusion because the models can’t disentangle the sociality of deer (implicit in the transmission parameterization) from any of the other rates you parameterized from literature or your expert elicitation.

I also don’t think the term environmental context correctly reflects what was the focus of the modeling scenarios. I suggest describing this as “epidemiological conditions” or “human and white-tailed deer interaction context”, or something similar to more precisely convey what varied among the scenarios.

Discussion sentence starting “We estimated sustained infections…” – What does infection risk refer to for this conclusion. I suggest using language that matches the analysis and result that characterizes this range of infection risk

Discussion sentence starting, “These results complement ongoing…” – provide some citations for these complementary efforts

Reviewer #2: My review is not included as an attachment:

SARS2 spillover in white-tailed deer in North America has been widespread and represents a complex challenge to our understanding of the disease ecology of the virus. The authors simulate the introduction, spread, and persistence of SARS-CoV-2 in white-tailed deer populations under four broadly generic scenarios including wild white-tailed deer and their captive conspecifics. These data are interesting and important for informing potential downstream control measures. The authors clearly explain the assumptions used in the modelling approach. The manuscript is well written and organized but is very long and should be reduced where/if possible. I only have minor comments/revisions:

Specific comments:

Lines 71-72: Authors should also include that zoonotic disease transmission is bi-directional (animal to human and human to animal).

Lines 143-145: This statement does not belong in the introduction. Please remove.

Lines 551-552: I think the authors meant to say “…average prevalence, probability of persistence, and incidence proportion….”

Figure 6: The way this figure is currently laid out makes the comparison across conditions hard. Would it be possible to put the conditions (e.g. wild rural, wild suburban etc) along the x-axis?

The line numbers end at 602, can the authors add these in when submitting the revised version of the manuscript?

**Have the authors made all data and (if applicable) computational code underlying the findings in their manuscript fully available?**

Reviewer #1: Yes

Reviewer #2: Yes

PLOS authors have the option to publish the peer review history of their article (what does this mean?). If published, this will include your full peer review and any attached files.

Reviewer #1: No

Reviewer #2: No
---

## [Decision Letter · Decision Letter 1]

2 May 2024

Dear Mr. Rosenblatt,

Thank you very much for submitting your manuscript "Epidemiological modeling of SARS-CoV-2 in white-tailed deer (Odocoileus virginianus) reveals conditions for introduction and widespread transmission" for consideration at PLOS Computational Biology. As with all papers reviewed by the journal, your manuscript was reviewed by members of the editorial board and by several independent reviewers. The reviewers appreciated the attention to an important topic. Based on the reviews, we are likely to accept this manuscript for publication, providing that you modify the manuscript according to the review recommendations.

Sincerely,

Benjamin Althouse

Academic Editor

PLOS Computational Biology

Rob De Boer

Section Editor

PLOS Computational Biology

Reviewer's Responses to Questions

**Comments to the Authors:**

Reviewer #1: The revisions are good and I think this manuscript is a well-done and valuable contribution with one exception remaining from the initial review:

In my view, the authors did not appropriately address this comment from the initial review: "Discussion sentence beginning “Our findings indicated…” [Line 651-653] You did not include any aspect of sociality. Some aspects were including in how you modeled transmission according to previous studies, but these were constant, as far as I could tell. Remove this conclusion because the models can’t disentangle the sociality of deer (implicit in the transmission parameterization) from any of the other rates you parameterized from literature or your expert elicitation."

This revised conclusion still implies that sociality was part of your findings. However sociality was not modeled as an explicit explanation to drive contact rates. The issue that remains is that this conclusion does not follow from your methods or results.

I have no doubts that white-tailed deer sociality contributes to transmission of SARS-CoV-2 and it would be acceptable to discuss or generate hypotheses about sociality influences transmission-relevant contact rates in white-tailed deer. Social structuring and interaction in white-tailed is a complex phenomena that has a quite a bit of accumulated literature- more on sociality than how sociality leads to contact and pathogen transmission- and one thing that is clear from this literature is that sociality is not a uniform (or low variance) phenomena in white-tailed deer.

I suggest removing the attribution of sociality as underpinning the proximity rates in this sentence. Such that is reads something like: Our findings indicated that epidemiological conditions and the proximity rates of white-tailed deer may lead to sustained transmission.

Then the authors could then chose to include a brief statement that social behaviors of white-tailed deer may (or are likely to) drive the proximity and contact rates or a longer discussion as they see fit.

**Have the authors made all data and (if applicable) computational code underlying the findings in their manuscript fully available?**

Reviewer #1: Yes

PLOS authors have the option to publish the peer review history of their article (what does this mean?). If published, this will include your full peer review and any attached files.

Reviewer #1: No

Figure Files:

Data Requirements:

Reproducibility:

References:

---

## [Decision Letter · Decision Letter 2]

18 Jun 2024

Dear Mr. Rosenblatt,

We are pleased to inform you that your manuscript 'Epidemiological modeling of SARS-CoV-2 in white-tailed deer (Odocoileus virginianus) reveals conditions for introduction and widespread transmission' has been provisionally accepted for publication in PLOS Computational Biology.

Best regards,

Benjamin Althouse

Academic Editor

PLOS Computational Biology

Rob De Boer

Section Editor

PLOS Computational Biology

Reviewer's Responses to Questions

**Comments to the Authors:**

Reviewer #1: Nothing further, the authors have adequately addressed my comments

**Have the authors made all data and (if applicable) computational code underlying the findings in their manuscript fully available?**

Reviewer #1: Yes

PLOS authors have the option to publish the peer review history of their article (what does this mean?). If published, this will include your full peer review and any attached files.

Reviewer #1: No

---

## [Editor Report · Acceptance letter]

5 Jul 2024

PCOMPBIOL-D-23-01712R2 

Epidemiological modeling of SARS-CoV-2 in white-tailed deer (Odocoileus virginianus) reveals conditions for introduction and widespread transmission

Dear Dr Rosenblatt,

I am pleased to inform you that your manuscript has been formally accepted for publication in PLOS Computational Biology. Your manuscript is now with our production department and you will be notified of the publication date in due course.

With kind regards,

Lilla Horvath
